# Integration of Microarray and Single-Cell RNA-Seq Data and Machine Learning Allows the Identification of Key Histone Modification Gene Changes in Spermatogonial Stem Cells

**DOI:** 10.3390/biology14040387

**Published:** 2025-04-08

**Authors:** Ali Shakeri Abroudi, Hossein Azizi, Melika Djamali, Ali Qorbanee, Thomas Skutella

**Affiliations:** 1Department of Cellular and Molecular Biology, Faculty of Advanced Science and Technology, Tehran Medical Sciences, Islamic Azad University, Tehran 1936893813, Iran; alishakeriabroudi@gmail.com; 2Faculty of Biotechnology, Amol University of Special Modern Technologies, Amol 4615664616, Iran; 3Department of Biology, Faculty of Science, Tehran University, Tehran 1417614411, Iran; mdjamali@ut.ac.ir; 4Department of Surgery, Faculty of General of Medicine, Koya University, Koya KOY45, Kurdistan Region—F.R., Iraq; ali.qurbany@koyauniversity.org; 5Institute for Anatomy and Cell Biology, Medical Faculty, University of Heidelberg, Im Neuenheimer Feld 307, 69120 Heidelberg, Germany; thomas.skutella@uni-heidelberg.de

**Keywords:** spermatogonia stem cell, microarray, bioinformatics, gene ontology, germ cell

## Abstract

Histone modifications help control gene activity and are important for maintaining the health of spermatogonial stem cells (SSCs), which are necessary for male fertility. In this study, we used advanced genetic analysis to identify key genes related to SSC function and aging. We discovered 2509 genes that exhibited differential expression in SSCs compared to other cells, with important genes including KDM5B, SCML2, SIN3A, and ASXL3 playing roles in how DNA is organized and read. Further analysis showed that these genes play essential roles in maintaining DNA structure and regulating gene activity. We also identified gene groups associated with SSC aging and discovered potential signals that help SSCs remain active and proliferate. These findings could contribute to developing new methods to protect male fertility and advance SSC research and treatments.

## 1. Introduction

Stem cells that can both self-renew and differentiate into other types of cells are essential for spermatogenesis, the process whereby males continue to produce sperm throughout their lives [1]. A complicated interaction between internal transcriptional programs and extrinsic signals from their milieu, usually known as the SSC niche, regulates these cells, which reside in the basal compartment of the seminiferous tubules [2]. Impairments in spermatogenesis, caused by disturbances in these delicately balanced processes, may lead to male infertility and disorders such as non-obstructive azoospermia [3]. Modifications to histones and other epigenetic factors are important regulators of SSC function. Covalent post-translational modifications to histone proteins, often known as “histone modification”, have the potential to drastically change the structure of chromatin and, by extension, gene expression. Chemical changes such as methylation, acetylation, phosphorylation, and ubiquitination regulate gene transcription by activating or repressing it. In the context of SSCs, histone modifications regulate the expression of genes which are essential for stem cell maintenance, proliferation, and differentiation [4].

The importance of histone-modifying enzymes in SSC control has been highlighted in recent research. For example, Hui et al. [5] showed that LSD1, a histone demethylase, is essential for SSC self-renewal as it keeps chromatin open, which enables the expression of important genes associated with stemness. Histone acetyltransferase KAT6B enhances SSC differentiation by acetylating histones at loci important in differentiation pathways, as revealed by Fisher et al. [6]. Based on these results, it seems that enzymes mediating certain histone modifications play a pivotal role in SSC biology [7]. Beyond regulating gene expression in SSCs, histone alterations also play important roles in chromatin remodeling, genome organization, and other activities [8,9]. A thorough epigenomic study of SSCs has been performed by Min et al. [10], who found that chromatin-remodeling complexes, such the SWI/SNF complex, are crucial in terms of allowing SSCs to properly differentiate into spermatocytes. According to their research, changes in transcriptional activity and chromatin accessibility were associated with distinct histone methylation patterns in SSCs, as opposed to other kinds of testicular cells [11].

Bioinformatics methods have also helped to identify important genes and pathways involved in SSC regulation, complementing these experimental investigations. Differently expressed genes (DEGs) in SSCs, in comparison to other cell types, such as fibroblasts, have been discovered thanks to high-throughput microarray and RNA-sequencing methods [12]. In their landmark work, Illi et al. conducted a comprehensive transcriptome investigation of mouse SSCs and found differently expressed histone-modifying enzymes, including KDM and SETD family members [13]. In addition, their research showed that SSCs retain their stem cell status by upregulating histone methyltransferases, leading to improved chromatin compaction and the suppression of genes linked to differentiation. Expanding upon these results, the current investigation seeks to provide further clarity regarding the function of histone alterations in human SSCs [14]. With a focus on histone-modifying enzymes, we compared the gene expression patterns of SSCs and fibroblasts using microarray analysis. Our goal was to determine which molecular and biological processes these enzymes controlled through building protein–protein interaction (PPI) networks and conducting gene ontology (GO) and KEGG pathway enrichment studies. This work adds to the existing literature on male infertility by combining data from the Gene Expression Omnibus (GEO) and The Cancer Genome Atlas (TCGA). As a result, we provide a thorough review of histone modification in SSCs and identify possible treatment targets in this area.

The areas of SSC biology and epigenetics have been investigated in several studies. In a comprehensive examination of the epigenetic landscape of SSCs, Gu et al. demonstrated that SSC differentiation dynamically regulates histone modifications such as H3K4me3 (an indication of active transcription) and H3K27me3 (indicating gene repression) [15]. The importance of polycomb repressive complexes (PRCs) in regulating SSC differentiation and self-renewal was also brought to light in this research. In addition, the histone demethylase KDM4D is essential for the epigenetic reprogramming that SSCs experience as they differentiate [16,17]. The cited authors’ research established a connection between spermatogenesis and histone demethylation, showing that KDM4D depletion halted SSC differentiation [18].

According to large-scale bioinformatic investigations, histone alterations govern important pathways and networks in SSCs. Important histone-modifying enzymes involved in SSC control have been identified in a systems biology study conducted by Chenarani et al. via integrating multi-omics data [19]. A number of genes, including KDM5A and CHD1, were identified as potential epigenetic regulators of SSC function as a result of their study of protein–protein interaction networks and gene regulatory circuits. Consistent with these results, in this investigation, we also found related DEGs and validated their functions in the maintenance of stem cells and processes related to the structure of chromatin. Collectively, our findings highlight the importance of further research into the epigenetic control of SSCs and the pivotal role played by histone alterations in SSC biology. This work advances our knowledge of the molecular pathways that control the preservation and differentiation of SSCs by building on these previous results; in particular, this new knowledge may have applications in the treatment of male infertility.

## 2. Materials and Methods

### 2.1. Design of the Experiment

This study was conducted from October 2016 to September 2017 using testicular samples obtained from three adult men, as in our previous research. Ethical clearance for human subject research was granted by two separate committees: a committee from the Amol University of Special Modern Technologies (Ir.asmt.rec.1403.06) and the Committee of the Medical Faculty of the University of Heidelberg (reference number S-376/2023). Prior to sample collection, all human volunteers were provided with detailed information, and written informed consent was obtained. The donors’ ages ranged from 23 to 67 years. The donor tissue was confirmed to be healthy and exhibited no pathological conditions (ages 23, 45, and 67) [20].

To investigate the characteristics of testicular adult stem cells, we compared gene expression patterns in two types of cultures: short-term (<2 weeks post-matrix-selection) spermatogonial stem cell (SSC) cultures and long-term (>2 months, up to 6 months) human adult germline stem cell (haGSC) cultures. These patterns were analyzed using microarray analysis and compared with gene expression profiles from human fibroblasts and human embryonic stem cells. This study involved three male subjects [20].

### 2.2. Cultivation of haGSCs

Following the removal of the tunica albuginea, the seminiferous tubules were carefully separated from the human testicular tissues. Enzymatic digestion was performed using collagenase type IV (750 U/mL; Sigma, Darmstadt, Germany), dispase II (0.25%; Roche, Basel, Switzerland), and DNase (5 μg/mL) to degrade the tubules in each sample. The digestion process was carried out at 37 °C for 30 min with gentle agitation in HBSS buffer containing Ca++ and Mg++ (PAA, Farnborough, UK) to obtain a single-cell suspension. To decelerate enzymatic digestion, 10% fetal bovine serum (FBS) (Thermo Fisher Scientific, Bremen, Germany) was added.

The most productive cell cultures were sub-cultured every two to three weeks in a 1:2 ratio. Maintaining optimal cell density in the culture wells was essential to prevent excessive cell dilution.

Donor Selection Criteria:

Eligible donors were adult men of reproductive age who were in good health and exhibited no signs of conditions that could cause infertility. Donors with a history of infertility, testicular cancer, or hereditary reproductive disorders were excluded to ensure that this study focused on normal SSC behavior. All the participants provided informed consent before the biological samples were collected.

Exclusion Criteria:

Donors with a history of cancer, mumps, cryptorchidism, or any infectious/systemic disease affecting SSCs were excluded. Individuals undergoing chemotherapy, hormone therapy, or other treatments affecting reproductive function were also not considered. Additionally, donors with smoking, alcohol, or drug abuse habits were excluded due to the potential impacts of such habits on sperm quality and SSC function [9,14,21].

### 2.3. hSSC Isolation

Spermatogonial cells were separated from the feeder layer or somatic cell monolayer on a culture plate by washing each sample in the culture medium. A single-cell suspension was prepared by gently re-suspending the cells before plating them on a 3.5 cm diameter culture plate. The cover of the dish was placed on a micromanipulation apparatus and positioned on the pre-heated (37 °C) working platform of a Zeiss inverted microscope.

The cells were meticulously collected using a pipette for micromanipulation at 20× magnification. After a short period in the culture, the distinctive morphology of spermatogonia became clearly visible. These cells were characterized by a high nucleus-to-cytoplasm ratio and a spherical shape with a diameter of approximately 6–12 μm. A bright cytoplasmic ring between the rounded nucleus and the outer cell membrane was often observed, facilitating their identification.

### 2.4. Collection of Single-Cell haGSC Colonies

Using enzymatic digestion, we separated typical haGSC and hESC colonies, along with a robustly growing hFibs colony, into individual cells. This allowed us to analyze the cells within the haGSC colony in greater detail. To investigate gene expression levels at a single-cell resolution, we employed a micromanipulation technique to manually isolate specific cells.

The primary objective of this approach was to obtain detailed information on the cellular properties of key genes associated with germline development and pluripotency. We aimed to assess gene expression variability among selected cells from a representative haGSC colony. Additionally, our goal was to cultivate colonies that exhibited germline- and pluripotency-associated gene expression patterns, as these were the most favorable for our study. Testicular tissues were collected from consenting adult male donors and immediately placed in ice-cold Hanks’ balanced salt solution (HBSS) supplemented with 1% penicillin/streptomycin to prevent contamination. The tunica albuginea was carefully removed using sterile forceps and a scalpel, and the seminiferous tubules were gently separated from the surrounding connective tissue. Enzymatic digestion was performed using collagenase type IV (750 U/mL), dispase II (0.25%), and DNase I (5 µg/mL) in HBSS buffer containing Ca++ and Mg++ at 37 °C for 30 min with gentle agitation. After digestion, the reaction was neutralized with 10% fetal bovine serum (FBS), and the sample was pipetted to create a single-cell suspension, which was filtered through a 40 µm cell strainer to remove debris. The cells were washed twice with PBS and centrifuged at 300× *g* for 5 min. Cell viability was assessed using a Trypan Blue exclusion test, and viable cells were plated on a gelatin-coated culture dish for further culturing and analysis.

### 2.5. Immunocytochemical Staining

Over the course of 24 plates, the cells were grown and treated with 4% paraformaldehyde. The samples were washed in PBS, permeabilized with 0.1% Triton in PBS, and then blocked with 1% BSA in PBS. After the blocking solution was removed, the cells were treated with primary antibodies. The procedure involved incubation with secondary antibodies unique to the species coupled with various fluorochromes after 30 rinses. For 5 min at room temperature, the cells were counterstained with 0.2 g/mL of 4′,6-diamidino-2-phenylindole (DAPI) before fixation with Mowiol 4–88 reagent. For all indications, the absence of primary antibodies in the sample served as a negative control. The tagged cells were examined using a confocal Zeiss LSM 700 microscope, and pictures were taken with a Zeiss LSM-TPMT camera (Zeiss LSM 880, Munich, Germany).

### 2.6. Microarray Analysis

As a negative control, RNA was extracted from testicular fibroblasts (hFibs), long-term haGSC cultures, short-term spermatogonia, and the hESC line H1 using the RNeasy Mini Kit (Qiagen, Hilden, Germany). The MessageAmp aRNA Kit (Ambion, Thermo Fisher Scientific, Germany) was then used to perform amplification. For every sample, 200 cells were collected per probe using micromanipulation equipment. Afterwards, 10 μL of RNA direct lysis solution was used to keep the cells at −80 °C. The microarray facility at Germany’s University of Tübingen Hospital was used to analyze the samples. An oligonucleotide array developed by Affymetrix called the Human U133 + 2.0 Genome was used to analyze gene expression. For biostatistical analysis and normalization, the raw data (CEL files) were sent to MicroDiscovery GmbH in Berlin, Germany.

### 2.7. Filtering of DEGs

To separate the SSCs from the healthy controls in the GSE10180 and GSE65194 datasets, the “limma” R program was used. After calculating the *p*-value for each DEG, the Bonferroni technique was used to make adjustments. A Bonferroni *p* < 0.01 and a |log fold change (FC)| ≥ 2 were used as thresholds to identify genes with substantially different expression levels compared to the control group, for which the levels were 2-fold higher.

### 2.8. KEGG Pathway and GO Enrichment Analyses

KEGG pathway analysis was used for the methodical analysis and annotation of gene activities. There are three categories into which the genes were placed via the GO database: “cellular component”, “biological process”, and “molecular function” (25). Using the “clusterProfiler” R package and a cut-off of *p* = 0.05, we performed KEGG pathway and GO enrichment analyses on the DEGs produced in the previous phase.

### 2.9. Integration of the PPI Network and Cluster Analysis

One biological database that may be used to forecast PPI pairings is STRING. Genes that had a total score greater than 0.9 were considered important DEGs after STRING was used to assess the interactions between DEGs. Afterwards, a PPI network of the important DEGs identified was constructed using Cytoscape (version 3.6.1; http://cytoscape.org/ accessed on 28 March 2025). The most essential modules of the PPI network were identified using the default settings of the Cytoscape plugin Molecular complex detection (MCODE).

### 2.10. Analysis of Clinically Significant Modules

To build the co-expression network, we used WGCNA, an R tool designed for weighted gene co-expression network creation. We used WGCNA’s automated one-step network construction and module detection method, employing its default parameters, such as the default minimal module size, a merge cut height of 0.25, an unsigned type of topological overlap matrix, and the calculation of Pearson’s correlation coefficients. To measure how comparable the co-expression of complete modules was, the first principal component calculation module eigengene (ME) was used. Potential associations between MEs and phenotypes were assessed by using Pearson’s correlation coefficients.

### 2.11. Validation of the Hub Genes Using scRNA-Seq Datasets

Previously generated 10× genomic datasets (GSE149512) were used for transcriptome analysis. The culture dataset and source materials provide an extensive background on how these datasets were created. As a consequence, human spermatogonia were used in both datasets. The “culture” scRNA-seq dataset is thought to have originated in spermatogonia-related research. Rather than affecting the spermatogonia population’s biology, the transgenes serve as markers for future studies. Using two separate sets of markers—CD9Bright/ID4-eGFP+—and fluorescence-activated cell sorting (FACS), the adult testis dataset was compiled.

### 2.12. Analyzing Cell–Cell Interaction Using CellChat

We used CellChat43 (version 1.6.1) with default settings to find notable ligand–receptor combinations so that we could investigate cell–cell interactions in various types of cells, such as non-malignant cells and malignant cells with varied cell-cycle sequences (CCSs). The CellChat framework smoothly incorporated the Seurat object that included the cell-type labels and count matrices for each individual cell. Then, to find the right ligand–receptor combinations, we used the CellChatDB database. We estimated communication probabilities and deduced the complex CellChat network by projecting receptors and ligands onto a protein–protein interaction (PPI) network. Additionally, we excluded cell types with fewer than 10 cells to make sure our results were accurate.

## 3. Results

### 3.1. HSSC Selection and Culture

Matrix selection (especially collagen non-binding/laminin binding) and CD49f-MACS were used to separate and concentrate spermatogonia from testicular samples of three adult male donors. Based on the positive results regarding DDX4 (VASA) and SSEA4 in early cultures, it is probable that spermatogonia were the most abundant cells in the defined populations. Regardless of the donor’s age or culture duration, the pure spermatogonia retained a consistent shape across all three samples. The cells had a spherical shape, a diameter of approximately 6–12 μm, and a high nucleus-to-cytoplasm ratio. A bright cytoplasmic ring between the round nucleus and the outer cell membrane was a telltale sign of this ratio. Spermatogonia were observed in all three cultures in different configurations, including pairs, chains, and small clusters linked by intercellular bridges. In addition to spermatogonia, larger cells (12–14 μm in diameter) with an oval shape and a lower nucleus-to-cytoplasm ratio were identified in the cultures. A significant reduction in levels of human testicular fibroblasts (htFibs) was noted in the unselected cell population, whereas the primary cultures of three samples showed a substantial expansion in the levels of htFibs in the non-selected fractions. Spermatogonial cultures without htFibs are depicted in Figure 1.

### 3.2. DEG Filtering in SSCs Versus Fibroblasts

From the expression profiles, 1256 DEGs were retrieved using the following criteria: |logFC| ≥ 1.5 and Bonferroni adjusted *p* < 0.01. To show how each gene was distributed based on the logFC and -log(*p*-value) values, scatter volcano plots were created (Figure 2A). Roughly 326 histone-modifying enzymes were analyzed using a microarray. The two datasets yielded 322 consistent DEGs when the comprehensive bioinformatics analysis was conducted (Figure 2). There was an increase of 19 genes and a decrease of 11 genes among those DEGs. The heat map also showed that the two datasets had the same pattern of gene expression (Figure 2B, Appendix A).

Through microarray analysis, we discerned three distinct types of stem cells (SSCs) and fibroblasts, which revealed 11 upregulated genes and 19 genes that were downregulated. The data shown in the figure are given here. Figure 2C,D show that microarray analysis of three SSC human samples revealed the following gene expression patterns: downregulation of *ASXL3, PBRM1, ATAD2, NAP1L4, STAG3, CSPP1, TSGA10, CHD1L, URI1, SYCP2, and KDM5B and upregulation of NCOA3, JARID2, STAG1, SCML2, SIN3A, TNP2, HIST1H4B, HIST1H1A SCML4, KDM5A, TSPY8, ATXN3L, H1FNT, MBD2, SAP130, EPC1, CHD2, HIRA,* and *KDM4D* (Appendix A).

### 3.3. PPI Network

To determine the interactions between DEGs, the online STRING database was used. Important DEGs were defined as genes with a total score greater than 0.9. The PPI network (Figure 3A) was built using a total of 244 edges and 95 important DEGs as nodes. Using the PPI network, MCODE was able to identify 28 genes and three major clusters (Figure 3B).

### 3.4. KEGG Pathway and GO Enrichment Analysis

KEGG pathway analysis was used to explore the DEGs’ roles. Figure 3 displays the findings that were considered the most significant for each functional category (Figure 3). BP was most abundant in the following pathways: GO:0031057, negative reg. of histone modification; GO:0006338, chromatin remodeling; GO:0031497, chromatin assembly; GO:0006325, chromatin organization; and GO:0051276, chromosomal organization. All of these are pathways involving up- or downregulated genes. The bulk of the genes that were either upregulated or downregulated were MF-enriched in activities related to histone H3-tri/di/monomethyl-lysine-4 demethylase, histone H3-methyl-lysine-4 demethylase, histone demethylase, and protein demethylase. Many genes that were up- or downregulated were concentrated in certain protein complexes, including those involving cohesin, lateral elements, Sin3, Sin3-type complexes, and PcG proteins (Figure 4, Appendix A).

### 3.5. Development of the WGCNA and Identification of Key Modules

The DEGs were classified into distinct modules using a WGCNA, in which the average linkage clustering approach was used to determine how similar their expression patterns were. In Figure 4A, three modules (MEblue, MEturquoise, and MEgrey) were identified and recognized using various colors. The dataset was then split into two phenotypes: one for SSCs and one for fibroblasts (normal). Figure 4B shows that the strongest association was between the blue ME module and the SSC phenotype. The most relevant genes for SSCs were identified from the 35 genes in the blue module, which included *ASXL3*, *PBRM1*, *ATAD2*, *NAP1L4*, *STAG3*, *CSPP1*, *TSGA10*, *CHD1L*, *STAG1*, *SCML2*, *SIN3A*, *TNP2*, *HIST1H4B*, *HIST1H1A SCML4*, *KDM5A*, *TSPY8*, *ATXN3L*, *H1FNT*, *MBD2*, *SAP130*, *EPC1*, *CHD2*, *HIRA*, and *KDM4D*, which showed upregulation (Figure 5).

### 3.6. Cell–Cell Interactions Involving Known Ligand–Receptor Interactions

After a number of cell types were identified, we evaluated possible linkages between all of the cell types in the tumor microenvironment. Approximately 1800 interactions that have been scientifically verified and supported were used. Intracellular matrix (ECM)–integrin connections, receptor tyrosine kinase (RTK) interactions, tumor necrosis factor (TNF) interactions, chemokine and cytokine interactions, and many other receptor–ligand families are all part of this network. Given the significance of B7 family members in histone-modifying enzyme genes, we also manually incorporated their identified interactions.

By searching for cases where different types of cells within the tumor microenvironment expressed both parts of a certain ligand–receptor interaction, we were able to find similar cell–cell interactions in each of the six syngeneic tumor models (Figure 6). The scoring of the ligand–receptor interactions involved taking the average expression of the receptor and multiplying it by the average expression of the ligand in the particular cell types under investigation. We used the average expression value for every cell type to reduce the possibility of false negatives caused by zero dropouts. By assigning a score to each tumor model, we were able to calculate the mean interaction score, allowing us to identify preserved linkages. Following the analysis of numerous cell–cell interactions (including 64 cell-type combinations and approximately 1500 ligand–receptor pairs translated to mouse homologs), we used a one-sided Wilcoxon rank-sum test and the Benjamini–Hochberg multiple hypothesis correction to determine the statistical significance of each interaction score. Although we examined interactions involving all known cell types, our main emphasis was on cases where the ligand was released by either macrophages or cancer-associated fibroblasts (CAFs). The fact that both cell types are major producers of different ligands led to this conclusion. In addition, we examined every possible relationship between genes that code for histone-modifying enzymes.

Finally, we found that numerous interactions were significantly correlated with one phenotype. Although different ligands were used, all of the interactions targeted the same receptor. The question of whether this relationship is due to a specific ligand or receptor, rather than physical contact itself, was raised by this finding. In order to test this theory, we calculated the Spearman correlations between SSC rate and receptor or ligand expression levels alone. According to our research, a high correlation with ligands or a significant connection with receptors is often associated with interaction scores that show a meaningful link with the phenotype. This result is not surprising, considering that both receptor and ligand expression levels influence interaction scores; the two variables are related and not exclusive of one another. Even though there was not a strong correlation between the receptor and the ligand, there were cases where the tumor growth rate was significantly correlated with the interaction score. This was especially true in the region of the plot where the expression of both ranged from −0.5 to 0.5. As a bonus, we noticed cases in the top-left and bottom-right corners of the image where there were clear and substantial connections between the expression of receptors and ligands. So, it is not only the correlations between ligands and receptors that are responsible for the link shown in the interaction scores (Figure 7).

## 4. Discussion

Enzymes that change histones and other parts of chromatin have become crucial players in controlling gene expression. Alterations such as methylation, acetylation, citrullination, phosphorylation, ubiquitination, and sumoylation may occur after histone proteins are translated [22]. Among the many enzymes involved in the dynamic regulation of histone modifications, lysine-specific histone methyltransferases (KMTs) are responsible for adding methylation marks to various proteins, while histone KDMs are responsible for removing them [23]. Many species, including humans, rely on histone methylation for transcription and genome stability [5,6]. Histone lysine residues may be mono-, di-, or trimethylated; the exact function of each mark is dependent on its location and the number of marks. Cancer and other physiological illnesses are therefore closely associated with chromatin-modifying enzyme dysregulation. When it comes to controlling cell fate, KDM5B is just as important as KDM5A. To govern cell lineages and the cell cycle, KDM5B removes H3K4me3 from the promoters of genes in mESCs [24]. One way that KDM5B helps with development is through keeping progenitors in an uncommitted state. For instance, mRNA levels of BMI1, a marker of neural cell lineage; Egr1, a marker of cell differentiation; and p27, an inhibitor of the cell cycle, are substantially increased when KDM5B is knocked down in mESCs [24]. The pluripotent state of undifferentiated stem cells is maintained by overexpressing KDM5B, which reduces the expression levels of these genes. Proper differentiation may be prevented by the constitutive expression of KDM5B in mESCs, a process that represses these genes. Despite the fact that KDM5A and KDM5B control cell-cycle genes throughout development, it seems that they have opposing functions in regulating the cell cycle during differentiation: while KDM5A encourages differentiation by boosting cell-cycle exit and progression, KDM5B increases stem cell proliferation by inducing cell-cycle exit. It is recommended that further research on KDM5A and research on KDM5B are carried out in tandem, as distinct cellular circumstances are taken into account when modulating the cell cycle during differentiation [25,26].

SCML2, a component of the polycomb repressive complex 1 (PRC1) that is particular to the germline, uses two separate but complementary methods to set up the male germline’s unique epigenome. In order to suppress somatic/progenitor genes on autosomes, SCML2 collaborates with PRC1 to enhance RNF2-dependent ubiquitination of H2A. Coordination of the proliferation and differentiation of male germ cells is required for the complicated process of spermatogenesis [27]. Little is known about the molecular mechanisms that govern this process, but we may assume that transcriptional regulation plays a role. In this work, the aim was to explore the role of chromatin-associated Sin3A in the formation of germ cell lineages in mice. Male germline Sin3A genetic inactivation causes sterility due to early and penetrant apoptotic death in Sin3A-deleted germ cells, a process that occurs at the same time as mitotic re-entry. The appearance of the Sertoli-cell-only phenotype in Sin3A-deleted testes is in line with the fact that Sin3A is absolutely necessary for the formation and/or survival of germ cells. Curiously, examination of transcripts showed that, when Sin3A is inactivated in germ cells, the expression programs of Sertoli cells change. These investigations have shown that the Sin3-HDAC complex plays an important part in the germ cell lineage of mammals and that there is a complicated transcriptional interaction between germ cells and their environment that helps mammals conceive [27,28]. The ASXL3 gene codes for a protein that is involved in controlling gene transcription via the presence of a zinc finger domain called a plant homeodomain (PHD). Specifically, the encoded protein binds to the nuclear hormone receptors oxysterols receptor LXR-alpha (LXRalpha) and thyroid hormone receptor-beta (TRbeta), thereby reducing their transcriptional activity and adversely regulating lipogenesis. It is also possible that the encoded protein blocks histone deubiquitination [29].

## 5. Conclusions

In conclusion, this study shed light on the pivotal role of histone modification genes in regulating spermatogonial stem cell (SSC) function and the aging process. Through the integration of microarray and single-cell RNA-sequencing data, we identified significant alterations in key histone modification genes such as KDM5B, SCML2, SIN3A, and ASXL3, which are involved in chromatin remodeling and gene expression regulation. Our protein–protein interaction network and gene ontology analysis revealed that processes such as chromatin organization and histone demethylation play crucial roles in maintaining SSC functionality. Moreover, our identification of gene co-expression modules and potential signaling pathways offers new insights into the mechanisms influencing SSC stemness and differentiation. These findings not only enhance our understanding of SSC aging but also highlight histone modification genes as potential therapeutic targets for male fertility preservation and the improvement of SSC culture methods. Future research should explore the therapeutic implications of these molecular targets, providing a pathway toward innovative fertility treatments for aging-related male infertility.

## Figures and Tables

**Figure 1 biology-14-00387-f001:**
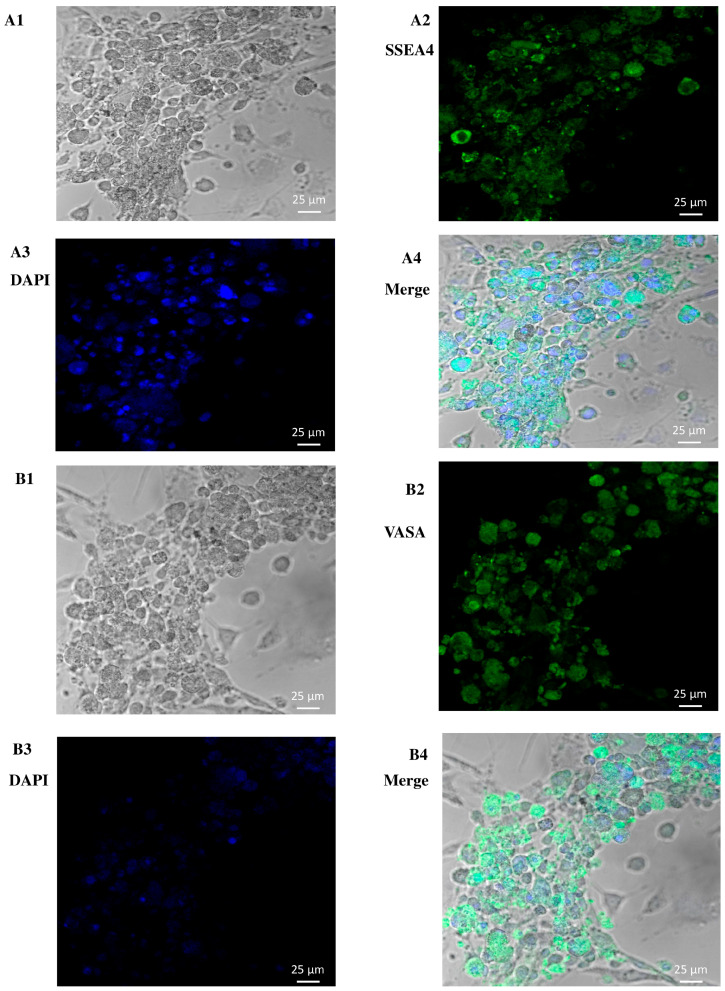
**In-vitro-cultivated human spermatogonia after matrix and CD49f selection.** During culturing, it was observed that the spermatogonia had the expected shape. Connected spermatogonia existed in all the cell cultures, either singly, in pairs, in chains, or in colonies. The cells were cultured for an extended duration using inactivated CF1 feeder cells. (**A1**) hSSC testicular cell expansion in the culture, (**A2**) SSEA4 expression in hSSCs, (**A3**) DAPI, (**A4**) merge, (**B1**) hSSC testicular cell expansion in the culture, (**B2**) VASA expression in hSSCs, (**B3**) DAPI, and (**B4**) merging. Scale bar: 25 μm.

**Figure 2 biology-14-00387-f002:**
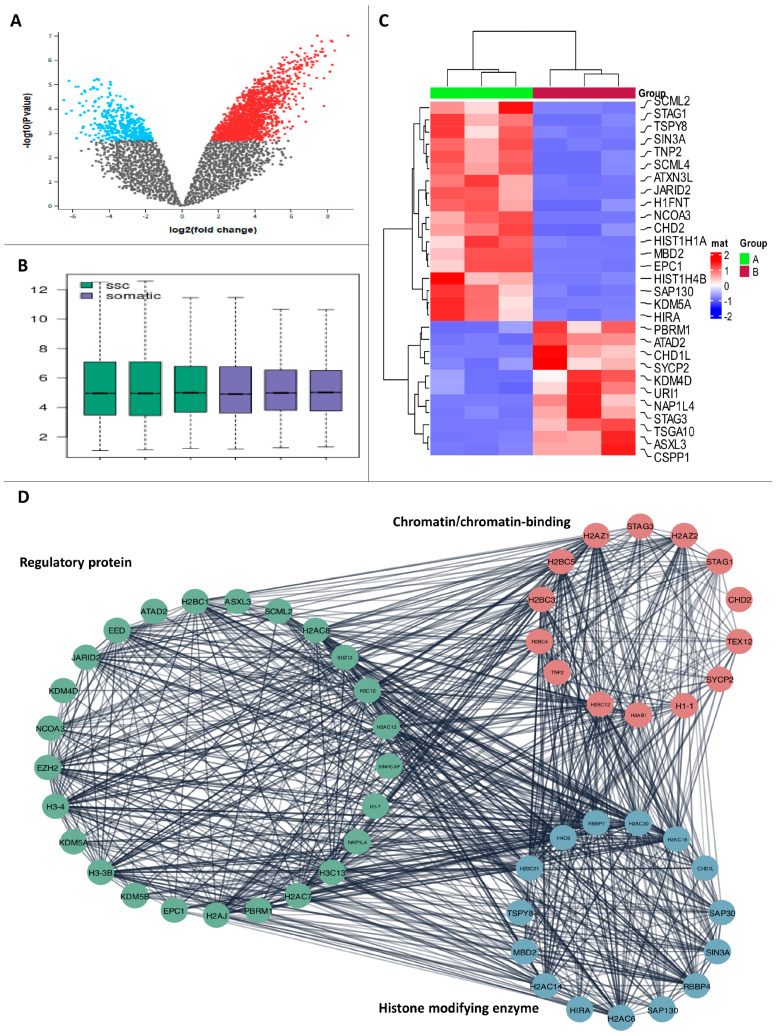
**Analysis of histone-modifying enzyme gene expression by microarray:** (**A**) volcano plot of differentially expressed genes based on microarray analysis, (**B**) correlation plot of SSCs and fibroblasts, (**C**) heatmap of histone-modifying enzyme DEGs, and (**D**) PPI of histone-modifying enzymes.

**Figure 3 biology-14-00387-f003:**
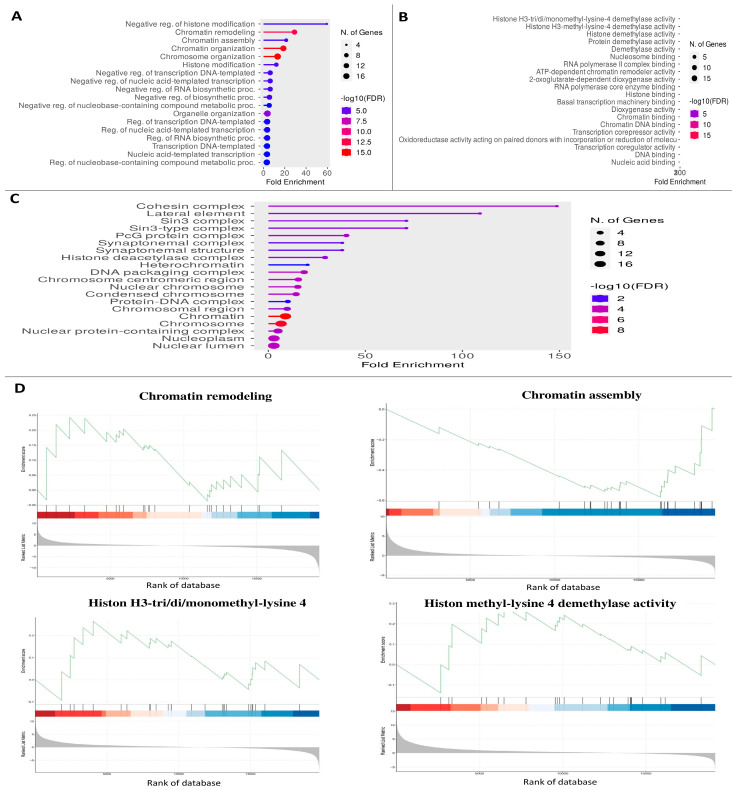
**Results of performing gene ontology (GO) enrichment analysis on the genes inside the module.** The colors correspond to the corrected *p*-values (BH), while the sizes of the dots correspond to the number of genes. This picture refers to (**A**) the biological processes, (**B**) molecular functions, (**C**) cellular components and signaling pathways, and signaling pathway analysis, (**D**) Signaling pathway analysis.

**Figure 4 biology-14-00387-f004:**
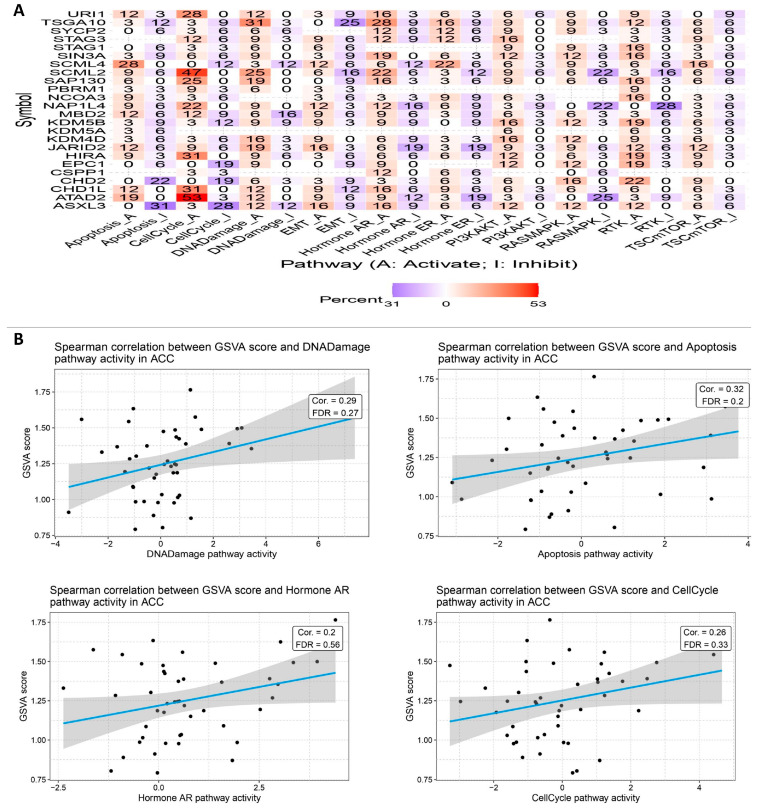
**Signaling pathways.** (**A**) Using the KEGG database, we investigated the key genes’ signaling pathways to find out how they contribute to signaling pathways. (**B**) When looking at the MCC scores of the genes, the red and yellow nodes reflect genes with high and low values, respectively.

**Figure 5 biology-14-00387-f005:**
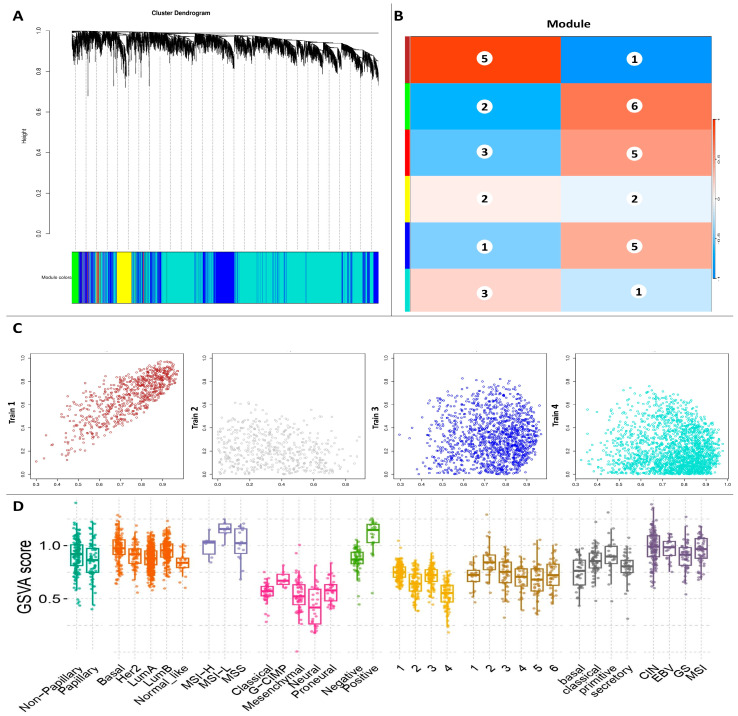
**WGCNA of the GEO datasets.** Clustering the DEGs from the GEO datasets yielded a gene dendrogram (**A**). Different colors are used to indicate a total of three modules: MEblue, MEturquoise, and MEgrey. The consensus MEs and phenotypes in the GEO datasets are explained in (**B**) correlation of WGCNA. (**C**) The gene dendrogram of the TCGA dataset, produced by clustering the DEGs. Differently colored markers are used to identify four separate modules: MEblue, MEbrown, MEgrey, and MEturquoise. (**D**) Phenotypic correlations from the TCGA dataset with respect to the consensus MEs. Abbreviations: differentially expressed gene (DEG), Gene Expression Omnibus (GEO), The Cancer Genome Atlas (TCGA), and module eigengene (ME).

**Figure 6 biology-14-00387-f006:**
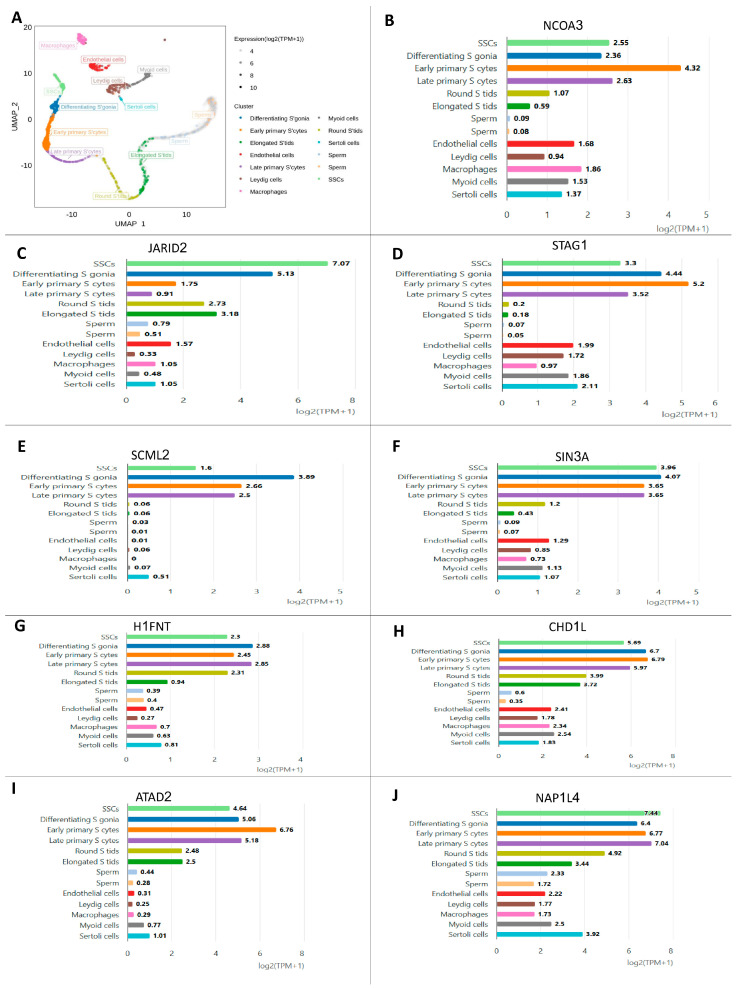
Single-cell RNA-sequencing investigation of human adult spermatogonia demonstrated histone-modifying enzyme genes similar to those seen in humans: (**A**) UMAP plot depiction of germ cells from merged single-cell RNA-sequencing data and (**B**) NCOR3, (**C**) JARID2, (**D**) STAG1, (**E**) SCML2, (**F**) SIN3A, (**G**) H1FNT, (**H**) CHD1L, (**I**) ATAD2, and (**J**) NAP1L4 gene expression in germ cells.

**Figure 7 biology-14-00387-f007:**
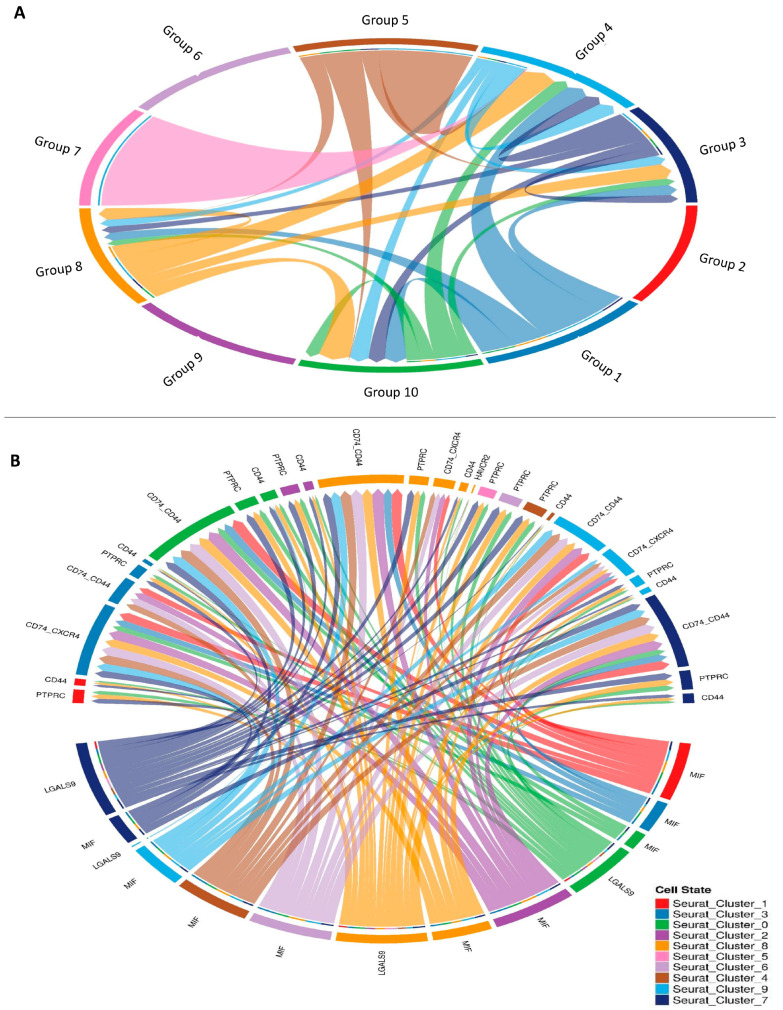
**Analysis of cellular communication:** (**A**) communication between individual cells that is influenced by genes that modify histones (Ligand–Receptor interactions 1–10 and their niche organization) and (**B**) communication between cells in a network that is influenced by the co-expression of genes that modify histones.

## Data Availability

The original contributions presented in this research are included in the article.

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
