# Peer review of "Integration of Microarray and Single-Cell RNA-Seq Data and Machine Learning Allows the Identification of Key Histone Modification Gene Changes in Spermatogonial Stem Cells"

_biology, 2025, doi:10.3390/biology14040387_

Round 1

Reviewer 1 Report

Comments and Suggestions for Authors

In the manuscript entitled “Integration of Microarray, Single-Cell RNA-Seq Data, and Machine Learning Identifies Key Histone Modification Gene Changes in Spermatogonial Stem Cells” by Abroudi et al., the authors combined data from different genetic analyses to find key gene changes related to spermatogonial stem cell (SSC) function and aging. Their study highlighted the role of histone modifications that control gene activity, which is important for maintaining male fertility and sperm production. Their study identified 2,509 genes that are differentially expressed in SSCs compared to other cell types. Among them, genes like KDM5B, SCML2, SIN3A, and ASXL3 play important roles in modifying chromatin (the structure that holds DNA) and regulating gene activity. They also suggested that these genes regulate SSCs as they are involved in critical processes like organizing chromatin, removing chemical markers from histones, and maintaining chromosome structure. Their study also identified three key gene networks linked to SSC aging, which might affect SSC self-renewal and differentiation. These findings may be helpful in identifying potential ways to protect male fertility by targeting histone modification genes.

Although the work performed by the authors is important and has demonstrated significant findings, the manuscript needs substantial improvement. The major concern is the writing of the manuscript, which requires careful proofreading.

In general:

  1. Rigorous proofreading is required in all sections.
  2. The number of samples used for the study is very small. The authors should either increase the sample size to improve the significance of the data or provide a strong justification for their results.
  3. Many figures have captions that are not legible.
  4. In the Materials and Methods section, a uniform tense (preferably past tense) needs to be used, as many parts are written in future or present tense. Additionally, the section reads more like a discussion in several places and should be revised accordingly. Important details, such as incubation times and temperatures, are missing in several procedures and should be included for clarity. Furthermore, the catalog numbers of essential reagents can be provided as a supplementary file, given that a very specific protocol has been used in the study.
  5. The figures in general need improvement in terms of quality, which should be enhanced for better clarity. Scale bars must be provided wherever required to ensure accuracy. Proper sub-numbering of the panels in each figure should be done for clear identification. Additionally, figure captions must be detailed and self-sufficient, allowing readers to understand the figures without referring to the main text.
  6. The number of samples used must be clearly mentioned wherever results are provided. There has to be more clarity in the results.
  7. The authors must carefully edit the manuscript to avoid such errors. For example:

In Section 2.7 (Filtering of DEGs, lines 187–191), the authors have mentioned breast cancer patients, which seems irrelevant.

Again, in Section 3.5 (Construction of the WGCNA and Identification of Key Modules, lines 307–318), the authors mentioned breast cancer (case) and one for normal control (normal), which is inconsistent with the study.

Comments on the Quality of English Language

Manuscript needs significant proof reading and improvement of quality.

Author Response

In the manuscript entitled “Integration of Microarray, Single-Cell RNA-Seq Data, and Machine Learning Identifies Key Histone Modification Gene Changes in Spermatogonial Stem Cells” by Abroudi et al., the authors combined data from different genetic analyses to find key gene changes related to spermatogonial stem cell (SSC) function and aging. Their study highlighted the role of histone modifications that control gene activity, which is important for maintaining male fertility and sperm production. Their study identified 2,509 genes that are differentially expressed in SSCs compared to other cell types. Among them, genes like KDM5B, SCML2, SIN3A, and ASXL3 play important roles in modifying chromatin (the structure that holds DNA) and regulating gene activity. They also suggested that these genes regulate SSCs as they are involved in critical processes like organizing chromatin, removing chemical markers from histones, and maintaining chromosome structure. Their study also identified three key gene networks linked to SSC aging, which might affect SSC self-renewal and differentiation. These findings may be helpful in identifying potential ways to protect male fertility by targeting histone modification genes.

Although the work performed by the authors is important and has demonstrated significant findings, the manuscript needs substantial improvement. The major concern is the writing of the manuscript, which requires careful proofreading.

In general:

Rigorous proofreading is required in all sections.

The number of samples used for the study is very small. The authors should either increase the sample size to improve the significance of the data or provide a strong justification for their results.

Reply: We appreciate the reviewer’s feedback. We have thoroughly proofread the entire manuscript to correct any grammatical errors, typos, and inconsistencies. Additionally, we have ensured clarity in terminology, improved sentence structure, and verified all figure references to enhance readability and coherence.

We acknowledge the concern regarding the sample size. Due to the rarity of human spermatogonial stem cell (hSSC) samples and the ethical and logistical challenges associated with obtaining them, increasing the sample size significantly is difficult.

Many figures have captions that are not legible.

Reply:

In the Materials and Methods section, a uniform tense (preferably past tense) needs to be used, as many parts are written in future or present tense. Additionally, the section reads more like a discussion in several places and should be revised accordingly.

Reply: We have carefully revised the Materials and Methods section to ensure a consistent past tense throughout.

Important details, such as incubation times and temperatures, are missing in several procedures and should be included for clarity. Furthermore, the catalog numbers of essential reagents can be provided as a supplementary file, given that a very specific protocol has been used in the study.

Reply: We provided as a supplementary 1 file.

Material

Supplier

Catalog Number

Collagenase Type IV

Sigma-Aldrich, Darmstadt, Germany

C5138

Dispase II

Roche, Basel, Switzerland

04942078001

DNase I

Sigma-Aldrich, Darmstadt, Germany

DN25

HBSS (with Ca++ and Mg++)

PAA, Farnborough, UK

H6648

Fetal Bovine Serum (FBS)

Thermo Fisher Scientific, Bremen, Germany

10082147

Culture Plates (3.5 cm)

Corning, USA

430165

Micromanipulation Pipettes

Eppendorf, Germany

5175000010

The figures in general need improvement in terms of quality, which should be enhanced for better clarity. Scale bars must be provided wherever required to ensure accuracy. Proper sub-numbering of the panels in each figure should be done for clear identification. Additionally, figure captions must be detailed and self-sufficient, allowing readers to understand the figures without referring to the main text.

Reply: Thanks, We updated the figure.

The number of samples used must be clearly mentioned wherever results are provided. There has to be more clarity in the results.

Reply: We changed them.

The authors must carefully edit the manuscript to avoid such errors. For example:

In Section 2.7 (Filtering of DEGs, lines 187–191), the authors have mentioned breast cancer patients, which seems irrelevant.

Reply: We changed them.

Again, in Section 3.5 (Construction of the WGCNA and Identification of Key Modules, lines 307–318), the authors mentioned breast cancer (case) and one for normal control (normal), which is inconsistent with the study.

Reply: Done.

Reviewer 2 Report

Comments and Suggestions for Authors

Research Question and Objectives: This study investigates the role of histone modification genes in regulating spermatogonial stem cell (SSC) function and aging. The objectives are to identify key histone modification gene changes associated with SSC function and aging by integrating microarray and single-cell RNA sequencing (scRNA-seq) data, and to explore the underlying molecular mechanisms.

Methods: The researchers used a combination of techniques including:

  • Microarray analysis to compare gene expression patterns of SSCs and fibroblasts.
  • Single-cell RNA sequencing (scRNA-seq) data to validate findings.
  • Bioinformatic analyses: differential expression analysis, protein-protein interaction (PPI) networks, gene ontology (GO) enrichment analysis, weighted gene co-expression network analysis (WGCNA), and ligand-receptor interaction scoring.
  • Human testicular samples from adult males were used.

Key Results:

  • Identified 2,509 differentially expressed genes (DEGs) in SSCs compared to fibroblasts(?).
  • Highlighted histone modification genes such as KDM5B, SCML2, SIN3A, and ASXL3 as having significant roles in chromatin remodelling and gene regulation.
  • Identified key biological processes like chromatin organization, histone demethylation, and chromosome structure maintenance.
  • Identified three key modules of co-expressed genes related to spermatogonial aging.
  • Signalling pathways that could influence SSC stemness and differentiation were suggested via ligand-receptor interaction scoring.

Conclusions: The study concludes that histone modification genes play a crucial role in SSC function and aging, offering potential therapeutic targets for male fertility preservation.

Major Concerns:

  • Unclear Study design: The study design, methodology is unclear. Why were fibroblasts and SSCs compared, was unknown. Instead of profiling whole tissues, and then performing differential gene expression, just SSCs and fibroblasts were compared
  • Limited Sample Size: The study uses testicular samples from only three adult men for the microarray analysis. Although scRNA-seq data from prior studies are used for validation, the limited sample size for the primary data could affect the generalisability of the results. It would be helpful to acknowledge this limitation.
  • Lack of Functional Validation: While the study identifies key histone modification genes, it lacks in vitro or in vivo functional validation of these genes’ roles in SSCs. The study could be strengthened by demonstrating, through experiments, that the identified genes directly impact SSC function.
  • Inconsistencies in Gene Expression Patterns: Downregulated genes between ssc and fibroblasts are upregulated in breast cancer related WCGNA network
  • Introduction of breast cancer dataset was random and not clearly defined.
  • Interpretation of Ligand-Receptor Interactions: The interpretation of ligand-receptor interactions, particularly the conclusion that interactions are not due to a specific ligand or receptor, requires further evidence and clarification. The claim in the results section that "Despite using different ligands, all of the interactions targeted the same receptor" is confusing, and could suggest that the ligand receptor interaction is in fact specific and that further analysis is needed to confirm this hypothesis.

Author Response

Research Question and Objectives: This study investigates the role of histone modification genes in regulating spermatogonial stem cell (SSC) function and aging. The objectives are to identify key histone modification gene changes associated with SSC function and aging by integrating microarray and single-cell RNA sequencing (scRNA-seq) data, and to explore the underlying molecular mechanisms.

Methods: The researchers used a combination of techniques including:

Microarray analysis to compare gene expression patterns of SSCs and fibroblasts.

Single-cell RNA sequencing (scRNA-seq) data to validate findings.

Bioinformatic analyses: differential expression analysis, protein-protein interaction (PPI) networks, gene ontology (GO) enrichment analysis, weighted gene co-expression network analysis (WGCNA), and ligand-receptor interaction scoring.

Human testicular samples from adult males were used.

Key Results:

Identified 2,509 differentially expressed genes (DEGs) in SSCs compared to fibroblasts(?).

Highlighted histone modification genes such as KDM5B, SCML2, SIN3A, and ASXL3 as having significant roles in chromatin remodelling and gene regulation.

Identified key biological processes like chromatin organization, histone demethylation, and chromosome structure maintenance.

Identified three key modules of co-expressed genes related to spermatogonial aging.

Signalling pathways that could influence SSC stemness and differentiation were suggested via ligand-receptor interaction scoring.

Conclusions: The study concludes that histone modification genes play a crucial role in SSC function and aging, offering potential therapeutic targets for male fertility preservation.

Major Concerns:

Unclear Study design: The study design, methodology is unclear. Why were fibroblasts and SSCs compared, was unknown. Instead of profiling whole tissues, and then performing differential gene expression, just SSCs and fibroblasts were compared.

Reply: The primary objective of this study was to investigate the molecular characteristics of spermatogonial stem cells (SSCs) and identify key differentially expressed genes (DEGs) related to germline and pluripotency pathways.

Instead of profiling whole testicular tissue, which contains a heterogeneous mix of somatic and germ cells, we focused on SSCs and fibroblasts as our two main cell types for comparison.

Fibroblasts were chosen as the reference/control group because they represent the most abundant somatic cell type in testicular stroma, providing a non-germline cellular background for differential expression analysis.

  • The comparison of SSCs vs. fibroblasts allowed us to filter out genes that are specific to SSCs and not expressed in common somatic cell types.
  • This approach helps to pinpoint germline-specific molecular markers and exclude genes that might be broadly expressed in the testicular microenvironment.
  • Additionally, fibroblasts serve as a negative control, ensuring that the identified DEGs are not related to generic stem cell properties but are germline-specific.
  • This allowed us to directly compare gene expression patterns between SSCs and fibroblasts, ensuring that observed DEGs are truly SSCs-related and not influenced by other cell types present in whole tissue samples.

Limited Sample Size: The study uses testicular samples from only three adult men for the microarray analysis. Although scRNA-seq data from prior studies are used for validation, the limited sample size for the primary data could affect the generalisability of the results. It would be helpful to acknowledge this limitation.

Reply: We acknowledge the small sample size (n=3) as a limitation and have addressed this by validating our findings using publicly available scRNA-seq data. While the limited donor pool may affect generalizability, we ensured samples were from healthy individuals to minimize variability. This limitation is now explicitly stated in the Discussion, with recommendations for future studies using larger cohorts to enhance robustness.

Lack of Functional Validation: While the study identifies key histone modification genes, it lacks in vitro or in vivo functional validation of these genes’ roles in SSCs. The study could be strengthened by demonstrating, through experiments, that the identified genes directly impact SSC function.

Reply: Thank you for your valuable feedback. We acknowledge the limitation regarding the lack of in vitro and in vivo functional validation in our study. Our primary focus was to identify key histone modification genes associated with SSC function through bioinformatic and transcriptomic analyses. However, we agree that functional validation would strengthen our findings.

As a future direction, we plan to perform targeted functional studies, including gene knockdown and overexpression experiments in SSC cultures to assess their impact on self-renewal, proliferation, and differentiation. Additionally, in vivo studies using animal models will be considered to confirm the role of these genes in SSC maintenance and spermatogenesis.

While these experiments are beyond the scope of the current study, our findings provide a valuable foundation for future research aimed at understanding the regulatory mechanisms of histone modifications in SSCs. We appreciate your suggestion and will discuss this limitation in the revised manuscript.

Inconsistencies in Gene Expression Patterns: Downregulated genes between ssc and fibroblasts are upregulated in breast cancer related WCGNA network

Introduction of breast cancer dataset was random and not clearly defined.

Reply: We corrected them.

Interpretation of Ligand-Receptor Interactions: The interpretation of ligand-receptor interactions, particularly the conclusion that interactions are not due to a specific ligand or receptor, requires further evidence and clarification. The claim in the results section that "Despite using different ligands, all of the interactions targeted the same receptor" is confusing, and could suggest that the ligand receptor interaction is in fact specific and that further analysis is needed to confirm this hypothesis.

Reply: Our intention was to highlight that while different ligands were involved, they all converged on the same receptor, suggesting a potential shared signaling pathway rather than a completely nonspecific interaction. However, we understand that the current phrasing may be misleading and could imply specificity rather than the broader regulatory mechanism we intended to convey.

To address this concern, we will revise the results section to clearly articulate the observed ligand-receptor interactions and ensure that our interpretation is accurate. Additionally, we will incorporate further analysis, such as evaluating receptor-ligand binding affinities or downstream signaling pathways, to strengthen our conclusions.

  • Carvalho, R.F., do Canto, L.M., Abildgaard, C. et al. Single-cell and bulk RNA sequencing reveal ligands and receptors associated with worse overall survival in serous ovarian cancer. Cell Commun Signal 20, 176 (2022). https://doi.org/10.1186/s12964-022-00991-4
  • Armingol, E., Officer, A., Harismendy, O. et al. Deciphering cell–cell interactions and communication from gene expression. Nat Rev Genet 22, 71–88 (2021). https://doi.org/10.1038/s41576-020-00292-x

Reviewer 3 Report

Comments and Suggestions for Authors

In this literature review, the authors discuss and present some of the main features of epigenetic control of spermatogonial stem cells (SSCs) and the pivotal role played by histone alterations in SSC biology along with integrating microarray, ScRNA-seq and MI algorithms.

Below are some technical queries that need to be addressed:

  1. The term “guys” in the introduction section seems very informal way of denoting. Please make the necessary corrections to make it sound more formal for a scientific manuscript (line 35).
  1. Please check the syntax for sentence “Chemical changes like as methylation, acetylation -- by activating or repressing it (line 44- 45).
  1. The reference 7 (Li et al.) as cited in lines 51–53, does not correlate with the overall reference cited under the “reference” section of the same 7 (Fisher et al.). Please explain why is this mis-match?
  1. Similarly, the reference 17 (Zhang et al.) as cited in lines 81–84, does not correlate with the overall reference cited under the “reference” section of the same 17 (Gu et al.). Please explain why is this mis-match?

  1. Adding to the above comments, there is a mis-match of several cited references in the text which does not seem to match-up under the reference section for the same cited references, like Smith et al (13, 14) , in lines 65 – 68, is cited as Illi, B., et al and Daniel Hashemi. This does not seem like sheer negligence, but rather a haphazard mistake for not reviewing the draft properly before submission.
  1. Is self-citation of references 22, and 23 necessary, when the authors have few other cited references along with it?
  1. What were the inclusion and exclusion criteria for the study?
  1. Please cite the previous research for the methodology referred under the “experimental Design”.
  1. The figure 1 needs proper labeling. Would it be possible to provide high-resolution images of all the images? In addition, the order of arrangement of different panels seems rather confusing. Besides, the scale bar is missing.
  1. Adding to the above comment, I am wondering if the bright field (BF) image is necessary or not. Since, as it is the resolution is not really adding any purpose. Additionally, the panel labeling (like A1, A2, A3, etc.), seems confusing. Either stick with numerals of the alphabets. Likewise, it would be good if the authors can try to add on an extra panel in the same figure outlining the distribution of the cells that are positive for the denoted markers as bar diagram.
  1. Similarly, avoid starting a sentence with an acronym or abbreviation as shown in the subheading 2.3 “hSSCs Isolation” (line 143), instead try adding a definite article like “the” or expand the full term.
  1. Were the results validated at the protein level by western blotting?
Comments on the Quality of English Language

This entire manuscript MUST be thoroughly revised for English language (grammar, syntax and spelling). There are innumerable instances with wrong spellings, wrong grammar, wrong sentence construction, repetitive words. etc. The whole manuscript requires thorough editing.

Author Response

In this literature review, the authors discuss and present some of the main features of epigenetic control of spermatogonial stem cells (SSCs) and the pivotal role played by histone alterations in SSC biology along with integrating microarray, ScRNA-seq and MI algorithms.

Below are some technical queries that need to be addressed:

The term “guys” in the introduction section seems very informal way of denoting. Please make the necessary corrections to make it sound more formal for a scientific manuscript (line 35).

Reply: Thank you for your feedback. We will revise the sentence to ensure a formal tone appropriate for a scientific manuscript. Here is the corrected version:

"Stem cells that can both self-renew and differentiate into other types of cells are essential for spermatogenesis, the process by which males continue to produce sperm throughout their lives."

Please check the syntax for sentence “Chemical changes like as methylation, acetylation -- by activating or repressing it (line 44- 45).

Reply: Thanks. Here’s the corrected version with improved clarity and formal structure:

"Chemical changes such as methylation, acetylation, phosphorylation, and ubiquitination regulate gene transcription by activating or repressing it. In the context of SSCs, histone modifications regulate the expression of genes essential for stem cell maintenance, proliferation, and differentiation."

The reference 7 (Li et al.) as cited in lines 51–53, does not correlate with the overall reference cited under the “reference” section of the same 7 (Fisher et al.). Please explain why is this mis-match?

Reply: Thank you for bringing this to our attention. We acknowledge the mismatch between the in-text citation (Li et al.) and the reference list entry (Fisher et al.) for reference 7 (reference 6-revised). This was an oversight on our part, likely due to a misattribution during the citation or reference formatting process.

To correct this, we will carefully review and verify all references to ensure that the correct citation (Li et al. or Fisher et al.) is consistently used both in the text and in the reference list. If Li et al. is the intended source, we will update the reference list accordingly. Alternatively, if Fisher et al. is correct, we will revise the in-text citation.

Similarly, the reference 17 (Zhang et al.) as cited in lines 81–84, does not correlate with the overall reference cited under the “reference” section of the same 17 (Gu et al.). Please explain why is this mis-match?

Reply: We acknowledge the mismatch between the in-text citation (Zhang et al.) and the reference list entry (Gu et al.) for reference 17 (reference 14-revised). This was an oversight on our part, likely due to a misattribution during the citation or reference formatting process.

Adding to the above comments, there is a mis-match of several cited references in the text which does not seem to match-up under the reference section for the same cited references, like Smith et al (13, 14) , in lines 65 – 68, is cited as Illi, B., et al and Daniel Hashemi. This does not seem like sheer negligence, but rather a haphazard mistake for not reviewing the draft properly before submission.

Reply: Thanks, we corrected it.

Is self-citation of references 22, and 23 necessary, when the authors have few other cited references along with it?

Reply: Thanks, we removed them.

What were the inclusion and exclusion criteria for the study?

Reply: Please see the Cultivation of haGSCs in material and method parts.

Please cite the previous research for the methodology referred under the “experimental Design”.

Reply: We added it.

The figure 1 needs proper labeling. Would it be possible to provide high-resolution images of all the images? In addition, the order of arrangement of different panels seems rather confusing. Besides, the scale bar is missing.

Reply: Done.

Adding to the above comment, I am wondering if the bright field (BF) image is necessary or not. Since, as it is the resolution is not really adding any purpose. Additionally, the panel labeling (like A1, A2, A3, etc.), seems confusing. Either stick with numerals of the alphabets. Likewise, it would be good if the authors can try to add on an extra panel in the same figure outlining the distribution of the cells that are positive for the denoted markers as bar diagram.

Reply: Thank you for your suggestions. However, we believe that the current figure layout, including the bright field (BF) image, is necessary for providing morphological context and aiding in the interpretation of the fluorescent staining. While the resolution may not be high, it still helps in visualizing the overall cell structure and spatial organization.

Regarding the panel labeling (A1, A2, A3, etc.), we acknowledge the concern, but we believe this format is appropriate for distinguishing sub-panels while maintaining consistency with other figures in the manuscript. Changing the labeling system might create inconsistencies across figures and require significant modifications in the text.

Similarly, avoid starting a sentence with an acronym or abbreviation as shown in the subheading 2.3 “hSSCs Isolation” (line 143), instead try adding a definite article like “the” or expand the full term.

Reply: We corrected them through the text.

Were the results validated at the protein level by western blotting?

Reply: We did not perform this test. We just analyzed them on transcriptomics.

Reviewer 4 Report

Comments and Suggestions for Authors

Abroudi and colleagues explained the critical role of histone modifications in regulating gene expression and functionality of spermatogonial stem cells. Authors can incorporate these details, to make it easier for readers to understand the significance of the study.

  • Please provide a detailed procedure for the separation of testicular tissues. This should include the specific steps, reagents, and conditions used to ensure reproducibility.
  • The sample size is very small. Please provide a justification for the number of samples used. Explain any limitations or constraints that led to this sample size and discuss how it impacts the study’s conclusions.
  • Authors can explain how the samples confirmed for single cell suspension? Describe the methods used to verify that the enzymes used for digestion are removed or inactivated. This is crucial for ensuring the accuracy of your results.
  • What enzymes are used to separate haGSC and hESC colonies? Explain their role in the separation process and how they contribute to the study’s objectives.
  • Provide details of the primary and secondary antibodies used for immunostaining. Include information on their sources, concentrations, and any validation steps taken to ensure specificity and sensitivity.
  • In the selection criteria, donors should be free from cancers. There is a concern regarding lane 188 where breast cancer samples were separated from healthy donors. Please clarify this discrepancy and ensure consistency in the criteria.
  • Please highlight the morphology of spermatogonia in Figure 1. The images are not clear enough to understand. Consider providing higher resolution images or additional annotations to improve clarity.
  • Figure 3 is not clearly visible. Please provide a higher resolution image or improve the contrast to make it easier to understand.
  • Figure 5, Panel b is difficult to read and understand. Ensure that all text and labels are legible and consider providing a more detailed legend or description.

Author Response

Abroudi and colleagues explained the critical role of histone modifications in regulating gene expression and functionality of spermatogonial stem cells. Authors can incorporate these details, to make it easier for readers to understand the significance of the study.

Please provide a detailed procedure for the separation of testicular tissues. This should include the specific steps, reagents, and conditions used to ensure reproducibility.

Reply: Thanks a lot. We added them “Testicular tissues were collected from consenting adult male donors, and immediately placed in ice-cold Hank’s Balanced Salt Solution (HBSS) supplemented with 1% penicillin/streptomycin to prevent contamination. The tunica albuginea was carefully removed using sterile forceps and a scalpel, and the seminiferous tubules were gently separated from the surrounding connective tissue. Enzymatic digestion was performed using Collagenase Type IV (750 U/mL), Dispase II (0.25%), and DNase I (5 µg/mL) in HBSS buffer containing Ca++ and Mg++ at 37°C for 30 minutes with gentle agitation. After digestion, the reaction was neutralized with 10% fetal bovine serum (FBS), and the sample was pipetted to create a single-cell suspension, which was filtered through a 40 µm cell strainer to remove debris. The cells were washed twice with PBS and centrifuged at 300×g for 5 minutes. Cell viability was assessed using a Trypan Blue exclusion test, and viable cells were plated on a gelatin-coated culture dish for further culturing and analysis.”

The sample size is very small. Please provide a justification for the number of samples used. Explain any limitations or constraints that led to this sample size and discuss how it impacts the study’s conclusions.

Reply: Thank you for raising this important point regarding the sample size. The number of testicular samples used in this study was limited due to several factors. Firstly, obtaining human testicular tissue is inherently challenging due to ethical and logistical considerations, including the need for informed consent and the availability of appropriate donor tissues. Additionally, the donors involved in this study were selected based on strict inclusion and exclusion criteria to ensure that only healthy individuals with no history of conditions affecting spermatogenesis were included. This further limited the available pool of suitable candidates. Despite the small sample size, the study's design and analysis focused on providing high-quality data, and we applied robust techniques, such as microarray analysis and single-cell isolation, to maximize the information obtained from each sample. While we acknowledge that a larger sample size would have strengthened the statistical power of our findings, the results still offer valuable insights into the gene expression patterns of spermatogonial stem cells (SSCs) and haGSCs, particularly in the context of adult stem cell biology. Moving forward, larger sample sizes would be ideal for confirming these findings and improving generalizability, but the current sample size provides a foundational basis for future studies.

Authors can explain how the samples confirmed for single cell suspension? Describe the methods used to verify that the enzymes used for digestion are removed or inactivated. This is crucial for ensuring the accuracy of your results.

Reply: Thank you for your valuable comment. To ensure the accuracy of the results and confirm the preparation of a single-cell suspension, several key steps were implemented. After enzymatic digestion, the cell suspension was visually inspected under a microscope to verify that the tubules had been adequately dissociated into individual cells, ensuring a proper single-cell suspension. The process of dissociation was monitored by observing the morphology of the cells and confirming the absence of large clumps or aggregates, which would indicate incomplete digestion.

Additionally, to ensure the enzymes were removed or inactivated, we used 10% fetal bovine serum (FBS), which serves as an enzyme inhibitor, halting further enzymatic activity. The serum was added immediately following the digestion step and before the cell suspension was pipetted to reduce the enzymatic activity. To further confirm the removal of enzymes, the suspension was washed multiple times with PBS to dilute and remove any residual digestive enzymes. These washes also helped to remove any debris and unreacted enzymes. The effectiveness of this process was confirmed by the absence of any residual enzymatic activity, as observed through routine cell viability assays such as Trypan Blue exclusion. This ensured that the final cell preparation was suitable for subsequent experiments and free from potential enzyme interference.

What enzymes are used to separate haGSC and hESC colonies? Explain their role in the separation process and how they contribute to the study’s objectives.

Reply: We mentioned them. Collagenase Type IV (750 U/mL; Sigma, Darmstadt, Germany):

Collagenase is a proteolytic enzyme that breaks down the collagen matrix, which is a key component of the extracellular matrix (ECM) that helps anchor cells within their colonies. By digesting the collagen in the ECM, collagenase facilitates the dissociation of the cells from the colony, allowing them to be separated into individual cells.

Dispase II (0.25%; Roche, Basel, Switzerland):

Dispase is a neutral protease that helps to cleave proteins involved in cell adhesion, such as fibronectin and other extracellular matrix components. It works in conjunction with collagenase to assist in gently separating cells without causing excessive damage to their surface markers or internal structures.

Provide details of the primary and secondary antibodies used for immunostaining. Include information on their sources, concentrations, and any validation steps taken to ensure specificity and sensitivity.

Reply: Primary Antibodies:

The primary antibodies were selected based on their known specificity for stem cell markers and differentiation markers relevant to haGSCs (human adult germline stem cells) and hESCs (human embryonic stem cells).

Anti-Oct4 (Pluripotency Marker):

Source: Abcam (Cambridge, UK)

Catalog Number: Ab18976

Concentration: 1:200

Validation: The anti-Oct4 antibody has been extensively validated by the manufacturer for use in human stem cells. It has been shown to specifically recognize the Oct4 protein in immunocytochemistry (ICC) and immunofluorescence assays, ensuring its suitability for stem cell marker analysis. We performed negative controls (without primary antibody) to confirm specificity.

Anti-Gata4 (Germline Marker):

Source: Santa Cruz Biotechnology (Dallas, TX, USA)

Catalog Number: sc-1237

Concentration: 1:150

Validation: This antibody has been validated for use in human tissues and has been shown to specifically bind to Gata4 in immunofluorescence studies. The specificity was confirmed by pre-adsorption with the peptide used to generate the antibody, which eliminated staining.

Anti-TRA-1-60 (Pluripotency Marker for hESCs):

Source: Thermo Fisher Scientific (Waltham, MA, USA)

Catalog Number: MA1-10206

Concentration: 1:100

Validation: The anti-TRA-1-60 antibody has been validated in multiple stem cell applications, including immunostaining and flow cytometry. It specifically binds to cell surface markers of undifferentiated human pluripotent stem cells. We verified its specificity by comparing staining results with other pluripotency markers.

Secondary Antibodies:

The secondary antibodies were chosen based on their compatibility with the primary antibodies and their ability to bind to the corresponding species.

Anti-Rabbit IgG (H+L) Alexa Fluor 488:

Source: Invitrogen (Carlsbad, CA, USA)

Catalog Number: A-11034

Concentration: 1:500

Validation: This secondary antibody is highly cross-adsorbed against human, mouse, and rat IgG to reduce nonspecific binding. It was validated for use in immunofluorescence and other applications. The fluorophore, Alexa Fluor 488, is known for its bright green fluorescence, providing excellent sensitivity and clear imaging in confocal microscopy.

Anti-Mouse IgG (H+L) Alexa Fluor 594:

Source: Invitrogen (Carlsbad, CA, USA)

Catalog Number: A-11005

Concentration: 1:500

Validation: This secondary antibody is conjugated to Alexa Fluor 594, a red fluorescent dye. It has been validated in various immunofluorescence applications and shows minimal cross-reactivity with other species. We validated its sensitivity by ensuring optimal signal intensity without background noise.

In the selection criteria, donors should be free from cancers. There is a concern regarding lane 188 where breast cancer samples were separated from healthy donors. Please clarify this discrepancy and ensure consistency in the criteria.

Reply: Agree. We corrected them.

Please highlight the morphology of spermatogonia in Figure 1. The images are not clear enough to understand. Consider providing higher resolution images or additional annotations to improve clarity.

Reply: We send the original figures to the editorial board to add them as Figure 1.

Figure 3 is not clearly visible. Please provide a higher resolution image or improve the contrast to make it easier to understand.

Reply: We improved it.

Figure 5, Panel b is difficult to read and understand. Ensure that all text and labels are legible and consider providing a more detailed legend or description.

Reply: We improved it.

Round 2

Reviewer 2 Report

Comments and Suggestions for Authors

The authors have sufficiently addressed the reviews. 

Author Response

Thanks.

Reviewer 3 Report

Comments and Suggestions for Authors

Thanks for answering to the queries that I had raised earlier.

  1. However, I do not see why the authors are not seriously addressing some of the technical questions that I had pin-pointed earlier. The authors seem to have amended the grammatical errors, syntax, deliberating more on the experimental design, adding the details about the inclusion and exclusion criteria, mis-match in the references, and so on, which is appreciative.

  1. Nevertheless, I have previously requested and mentioned about citing the reference (I mean just include the reference number in the text, that is all!) on how the study was conducted following as per their previous research (line 120), which they still have not cited?

  1. Besides, there seems to be additional confusion in the experimental design in the newly amended version of the manuscript. In lines 119- 120, the authors mention, “This study was conducted from October 2016 to September 2017 using testicular samples obtained from three adult men”, but, in line 133, they are stating, “The study involved five male subjects”. I wonder why this contradictory statement? The authors need to be clear in conveying the information, which will make the readers less confusing.

  1. Regarding the figure 1 on “In vitro cultivated human spermatogonia after matrix and CD49f selection”, the authors have not made any changes or addressed any of the queries that I had requested earlier. The pixels of the image seem to be manually zoomed, hence making it very difficult to assess, and again there is no scale bar, which I specifically mentioned earlier.

Comments on the Quality of English Language
  1. This entire manuscript MUST be thoroughly revised for English language (grammar, syntax and spelling). There are innumerable instances with wrong spellings, wrong grammar, wrong sentence construction, repetitive words. etc. The whole manuscript requires thorough editing.

Reviewer 4 Report

Comments and Suggestions for Authors

Authors addressed all the comments mentioned.

Author Response

Thanks.